# Changes in global groundwater organic carbon driven by climate change and urbanization

Liza K. McDonough[1,2] ✉, Isaac R. Santos [3,4], Martin S. Andersen[1,5], Denis M. O'Carroll[1,5], Helen Rutlidge[1,5], Karina Meredith[6], Phetdala Oudone [1,2], John Bridgeman[7], Daren C. Gooddy [8], James P.R. Sorensen[8], Dan J. Lapworth[8], Alan M. MacDonald [9], Jade Ward[8] & Andy Baker [1,2]

Climate change and urbanization can increase pressures on groundwater resources, but little is known about how groundwater quality will change. Here, we use a global synthesis ($n =$ 9,404) to reveal the drivers of dissolved organic carbon (DOC), which is an important component of water chemistry and substrate for microorganisms that control biogeochemical reactions. Dissolved inorganic chemistry, local climate and land use explained ~ 31% of observed variability in groundwater DOC, whilst aquifer age explained an additional 16%. We identify a 19% increase in DOC associated with urban land cover. We predict major groundwater DOC increases following changes in precipitation and temperature in key areas relying on groundwater. Climate change and conversion of natural or agricultural areas to urban areas will decrease groundwater quality and increase water treatment costs, compounding existing constraints on groundwater resources.

[1] Connected Waters Initiative Research Centre, UNSW Sydney, Sydney, NSW 2052, Australia. [2] School of Biological, Earth and Environmental Sciences, UNSW Sydney, Sydney, NSW 2052, Australia. [3] National Marine Science Centre, Southern Cross University, Coffs Harbour, NSW 2450, Australia. [4] Department of Marine Sciences, University of Gothenburg, 40530 Gothenburg, Sweden. [5] School of Civil and Environmental Engineering, UNSW Sydney, Sydney, NSW 2052, Australia. [6] Australian Nuclear Science and Technology Organisation (ANSTO), New Illawarra Rd, Lucas Heights, NSW 2234, Australia. [7] University of Bradford, Bradford, West Yorkshire BD7 1DP, UK. [8] British Geological Survey, Maclean Building, Wallingford OX10 8BB, UK. [9] British Geological Survey, Lyell Centre, Research Avenue South, Edinburgh EH14 6AJ, UK. ✉email: lizam@ansto.gov.au

Groundwater is the largest global source of fresh water. The potability of groundwater is highly dependent upon the concentration of dissolved organic carbon (DOC) due to its ability to alter water chemistry and microbial abundances[1–6]. Over 100,000 lifetime cancer cases in the United States (US) can be attributed to contaminants in drinking water. A large proportion of the risk identified is associated with the presence of disinfection by products (DBPs) and arsenic[7], both of which are strongly linked to DOC[3,8–11]. Chlorination and ozonation used for water treatment can result in harmful by-products including 3-chloro-4-dichloromethyl-5-hydroxy-2(5 H)-furanone, brominated acetic acid, trihalomethanes (THMs), formaldehyde, halogenated acetic acids, due to the presence of organic matter[12]. These by-products can be genotoxic, carcinogenic or result in tumors[12]. Since most of the health impacts caused by dissolved organic matter (DOM) are related to the formation of by-products and depend on the concentrations of other water chemical parameters, the World Health Organization[12] and many countries including Australia[13] do not regulate total organic carbon (TOC) or DOC concentrations in drinking water directly. Countries such as US[14], Canada[15], France[16], China[17] and South Africa[18] highlight potential concerns related to THM formation, health effects and esthetic quality in the broad DOC range of $0-5$ mg $L^{-1}$ during treatment.

In addition to health and esthetic impacts, the presence of DOC in water can lead to membrane fouling after ozonation. In order to avoid fouling and remove DOC, a biological filtration step is advised to be added to the water treatment process for water containing DOC concentrations >1 mg $L^{-1}$ [19]. This indicates that even relatively small DOC increases in raw groundwaters can have impacts not only on human health and water esthetics, but also on the ease and cost of water treatment. High DOC concentrations can also increase the mobility of other contaminants in groundwater, including heavy metals and nutrients, by complex association with dissolved or colloidal organic matter (OM)[3].

Climate variables such as temperature and precipitation impact on net primary production and microbial activity in ecosystems[20,21]. This drives availability of vegetation and its decomposition to DOC[21,22]. Changed precipitation, increasing temperatures and evaporation rates and patterns under future climate change scenarios are expected to alter biomass, impact surface water quantity[23], and subsequently increase domestic and agricultural reliance on groundwater resources. Increasing reliance on groundwater due to climate change impacts may be compounded by urbanization and global population growth which may increase contamination[24]. Recent research has focused on how climate change and urbanization will change groundwater quantities[25,26], however understanding the impact of climate change and urbanization on the quality of freshwater resources is also important[23,27]. Establishing links between climate change and groundwater quality requires large datasets to produce meaningful global estimates.

Here, we quantify the change in groundwater DOC related to climate change and urban land cover. We present the largest global dataset of 9,404 published and unpublished groundwater DOC concentrations (Supplementary Table 1) obtained from aquifers in 32 countries across 6 continents (Fig. 1). We provide an analysis of global groundwater DOC concentrations and quantify its key drivers. Specifically, we forecast changes in DOC concentrations due to projected changes in temperature and precipitation, as well as potential increases as a result of urban land use.

## Results and discussion
**Global groundwater DOC.** Groundwater DOC concentrations vary spatially and are usually lower than surface water concentrations. The global mean, median and standard deviation of groundwater DOC concentrations are 3.8, 1.2, and 14.8 mg C $L^{-1}$ respectively (Fig. 1). Most groundwater DOC concentrations fall within the 0–5 mg C $L^{-1}$ range, with 84.1% of samples <5 mg C $L^{-1}$ (Fig. 1a), with the dataset dominated by countries in low and mid latitudes.

Variations in DOC concentrations between countries (Fig. 1) are likely to be related to recharge rates and aquifer types. World-wide Hydrological Mapping and Assessment Programme (WHY-MAP) data[28] suggests that within the US dataset (Supplementary Fig. 1 [29]), major groundwater basins contain significantly lower DOC concentrations than local and shallow aquifers, and complex hydrogeological structures (both $p < 2.2 \times 10^{-16}$, Supplementary Fig. 2). Therefore, groundwater age and depth seem to control groundwater DOC. There are also significantly higher DOC concentrations identified in aquifers with <100 mm year$^{-1}$ recharge compared to those with high recharge rates (100–300 mm year$^{-1}$, $p = 2.342 \times 10^{-7}$) and very high recharge rates (>300 mm year$^{-1}$, $p = 4.857 \times 10^{-5}$, Supplementary Fig. 2), which could indicate a dilution effect.

**Groundwater DOC controls.** To determine the drivers of global DOC concentrations in groundwater, we generated a linear mixed model (Supplementary Table 2) for a large dataset ($n = 2196$) collected by the National Water Quality Assessment (NWQA) program of the US Geological Survey (USGS)[29]. This dataset was selected because it contained reliable data on chemical parameters unavailable for other samples. This allowed us to extract supplementary climatic data[30] (Supplementary Table 3), water table depth[31] and land use data[32,33] for analysis in the model.

Overall, the model explained 47.7% of the variation in DOC concentrations, with 31.3% explained by the fixed factors alone (all fixed and random factors), and 16.3% explained by the random factor aquifer age (age of host rock). Our analysis (Supplementary Fig. 3, Supplementary Table 2) shows positive correlations between DOC and temperature in the wettest quarter of the year ($p < 2 \times 10^{-16}$), groundwater temperature ($p < 2 \times 10^{-16}$), and dissolved calcium (Ca) ($p < 2 \times 10^{-16}$), potassium (K) ($p = 2 \times 10^{-13}$) and iron (Fe) ($p < 2 \times 10^{-16}$, Supplementary Fig. 4). There was also a weaker relationship between DOC and manganese (Mn) ($p < 0.039$). We also found negative relationships between DOC and temperature in the warmest quarter of the year ($p < 2 \times 10^{-16}$), precipitation in the driest month of the year ($p = 0.001$), silica (Si) ($p = 2 \times 10^{-6}$), pH ($p = 4.06 \times 10^{-5}$), sample depth below land surface ($p < 2 \times 10^{-16}$), land elevation ($p = 1 \times 10^{-6}$) and dissolved oxygen (DO) ($p < 2 \times 10^{-16}$, Supplementary Fig. 5). Our analysis revealed negative relationships ($p < 0.01$) between DOC and sodium (Na) ($p = 0.001$), and DOC and precipitation in the wettest month of the year ($p = 0.001$). Areas of urban land use were identified as having 19% higher groundwater DOC concentrations than natural or agricultural areas. Water table depth as a variable improved the overall model fit but was not a significant predictor of DOC ($p = 0.071$). The factors correlated with decreased and increased groundwater DOC concentrations are presented in Fig. 2. While the model represents large scale relationships between DOC and control variables, these relationships can vary locally due to site specific factors. Our model implies that large scale groundwater DOC concentrations are determined by the interaction of four major controlling factors. These include climate, urban land-use, water chemistry (redox controls), and aquifer age and groundwater residence times (Fig. 3).

**Climate controls.** Temperature and precipitation play an important role in predicting groundwater DOC concentrations.

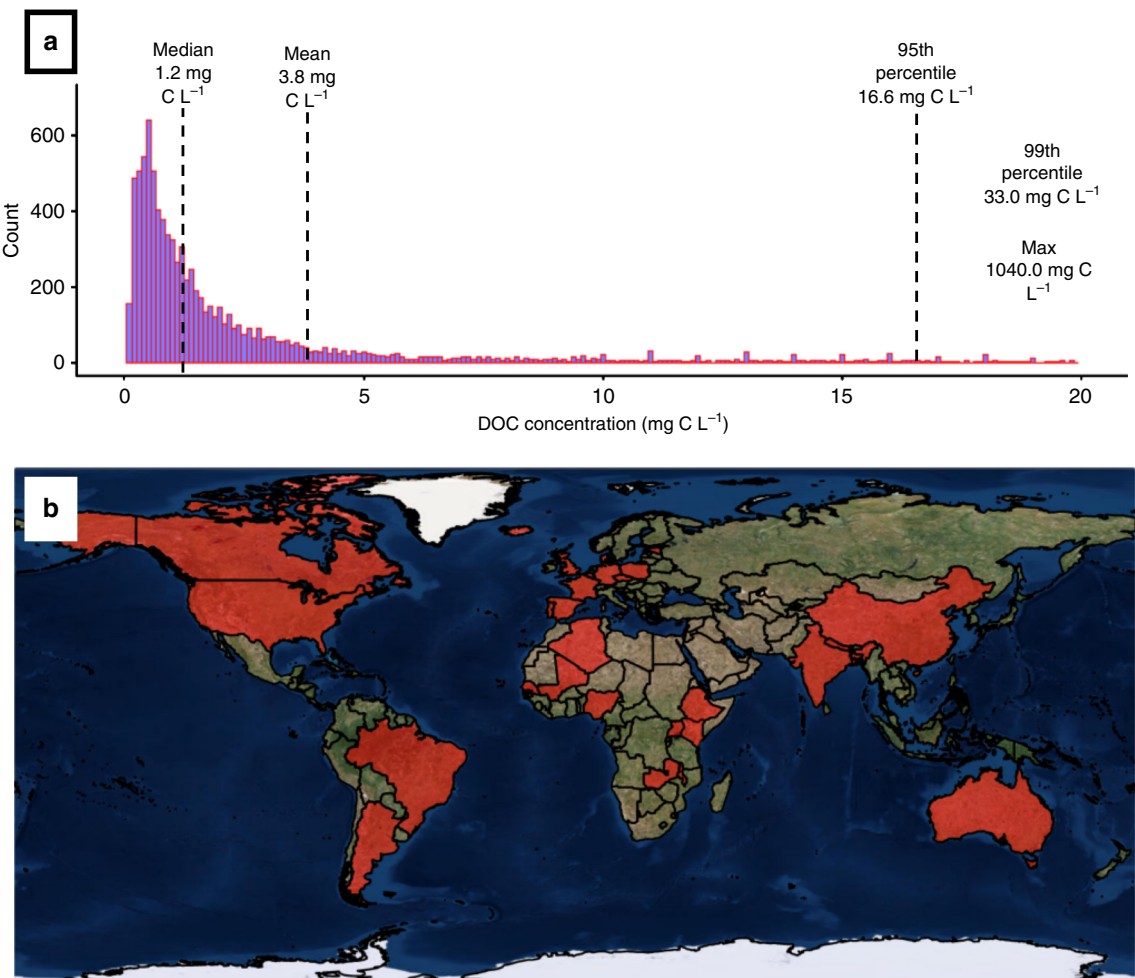

**Fig. 1 Global groundwater DOC concentrations. a** Histogram showing global groundwater DOC concentrations (mg C L$^{-1}$). Sample sizes for individual countries ranged from 5 to 5,812, with 14 out of 32 countries having $n < 30$. We have therefore presented aggregated data. Samples above 20 mg C L$^{-1}$ are not included in the graph for clarity ($n = 337$). The black dashed lines indicate the global median (1.2 mg C L$^{-1}$) and mean (3.8 mg C L$^{-1}$) and 95th percentile (16.6 mg C L$^{-1}$) values, respectively. Also shown are the 99th percentile value and the maximum value (33.0 and 1040.0 mg C L$^{-1}$, respectively). **b** Countries from which groundwater DOC data were obtained are shown in red.

Overall, DOC decreases by $9.5 \pm 1.1\%$ for every 10 mm increase in precipitation in the driest month of the year and decreases by $2.5 \pm 0.8\%$ for every 10 mm increase in precipitation in the wettest month of the year. This is likely due to a dilution effect whereby accumulated soil DOM infiltrates the aquifer during initial rainfall and is later diluted by additional rainfall[34]. In arid climates, some of these trends may be reversed (Fig. 3). For example, a decrease in aridity represented by decreased temperature and increased precipitation, would increase groundwater DOC concentration since the precipitation in the wettest month of the year is not enough to cause significant dilution. Furthermore, in arid climates, groundwater DOC concentrations would be low due to the high temperatures and low rainfall reducing vegetation cover and bioavailable DOM[35]. We observed these trend reversals in linear analyses for the smaller ($n = 79$ after removing incomplete data, Supplementary Fig. 6) Australian dataset (Supplementary Table 4, Supplementary Table 6, Supplementary Note 1).

The model shows an overall groundwater DOC concentration increase by $3.4 \pm 0.3\%$ for every $1\,°C$ increase in average air temperatures in the wettest quarter of the year and $4.6 \pm 0.5\%$ for every $1\,°C$ increase in sample groundwater temperature. In contrast, groundwater DOC concentrations decrease by $8.9 \pm 1.1\%$ for every $1\,°C$ increase in temperatures in the warmest quarter of the year. The source of DOC is dependent upon

availability of water. In humid climates, increases in mean surface temperature in the wettest quarter of the year and increased groundwater temperatures are likely to cause increased temperatures in the soil zone. Under conditions of increased soil moisture, warm temperatures can stimulate biological activity, DOM priming[36], and an increase in groundwater DOC.

**Water chemistry.** In the saturated zone, redox conditions and pH are strongly related to DOC concentration, with DOC concentrations $9.2 \pm 2.4\%$ lower for each unit increase in pH, and $6.8 \pm 0.6\%$ lower with every 1 mg L$^{-1}$ increase in DO. We also observe a $4.5 \pm 0.4\%$ increase in DOC associated with a 10 mg L$^{-1}$ increase in Ca. The smaller Australian dataset ($n = 79$) was consistent with the larger US dataset ($n = 2916$) (Supplementary Table 4). The mineralization of DOC consumes DO, produces $CO_2$ and subsequently decreases pH, resulting in calcite dissolution and dissolved Ca production. Once conditions become anoxic and biodegradation rates reduce, pH levels increase. The relationship between DOC and Ca, as well as microbial respiration by-products such as ammonium has been observed in regional-scale studies[3].

We also show that reduced dissolved species of Fe(II) and Mn (II) are positively correlated with DOC concentrations. This trend

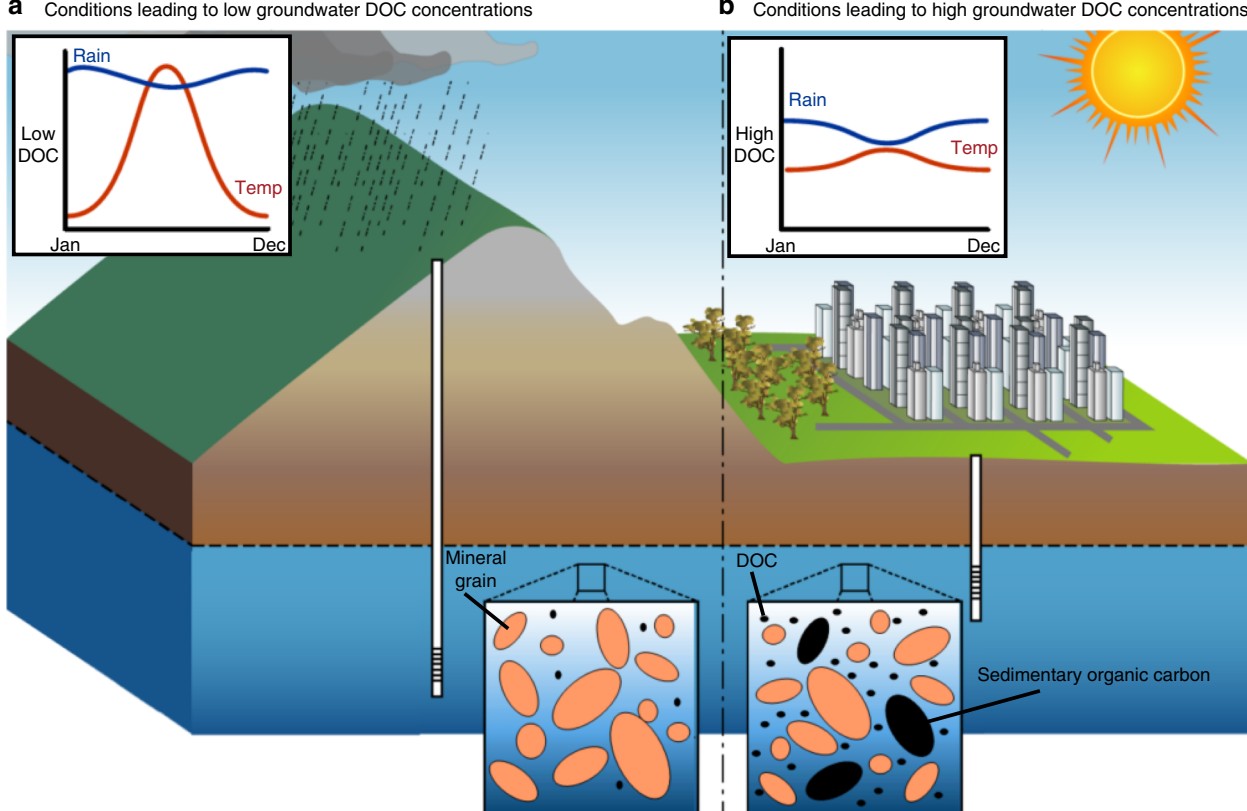

**Fig. 2 Factors and processes leading to low and high groundwater DOC concentrations. a** Conditions leading to low groundwater DOC concentrations.
**b** Conditions leading to high groundwater DOC concentrations. Factors negatively correlated with groundwater DOC concentrations include increasing pH,
DO, Na and Si, precipitation in the wettest and driest months, temperature in the warmest quarter, sample depth, elevation and aquifer age. Factors
positively correlated with groundwater DOC concentrations include Mn, Ca, Fe, and K, groundwater temperature and temperatures in the wettest quarter.
Urban land use was found to be 18% and 19% higher in groundwater DOC concentrations than agricultural and natural land uses, respectively.

may be explained by microbial use of Mn-oxides and Fe-oxides as
alternative electron acceptors to DO in the presence of DOC
under anoxic conditions[37]. Aerobic microbes metabolize carbon
at a faster rate than anaerobic microbes, therefore a lack of DO
limits DOC biodegradation. Fe can accumulate within oxic
sediment layers due to oxidation and precipitation of dissolved Fe
in young sediments[38]. This Fe can become coated with OM and
re-dissolve under reduced conditions releasing the OM, which
increases both Fe and DOC. Decreased sulfate ($SO_4$) and chloride
(Cl) deposition due to recent emission regulations increases DOC
concentrations in surface waters[39] which could also lead to
increased DOC concentrations in shallow groundwaters.

**Aquifer age and groundwater evolution.** The age of the geo-
logical formation, or aquifer age, explained 16.3% of variability in
groundwater DOC. Groundwater in younger aquifers of Cenozoic
sediments contained 41% higher DOC concentrations than older
Mesozoic and Paleozoic Era aquifers which support previous
observations in smaller datasets[4]. Despite site specific observa-
tions of high groundwater DOC associated with older aquifers[40],
the model suggests sedimentary OM in young aquifers is more
likely to be mobilized than in older, lithified aquifers. Other
studies have also reported higher groundwater DOC concentra-
tions originating from the matrix of younger aquifers[41,42].

We also observed a decrease in groundwater DOC concentra-
tions of 7.7 ± 0.6% for every 10 m increase in sample depth. As
DOC moves through porous media it undergoes filtration and
oxidation to DIC. Consolidated sediment pore-throat sizes can
occur in sizes much smaller than DOC, which is defined as the

fraction of total organic carbon passing a membrane with pores
between 0.2 to 0.7 μm. For example some pore-throat sizes in
Permo-Triassic sandstones have been determined to be as low as
0.01 μm[43]. In addition, deeper groundwaters often have longer
residence times[44]. This is implied by a positive relationship
between the mineral weathering product Si and sample depth
($p < 2 \times 10^{-06}$). We found a negative correlation between
groundwater DOC and Si, with DOC decreasing by 6.3 ± 1.3%
with every 10 mg L$^{-1}$ increase in Si. This relationship has also
been observed in surface waters in the US[45]. The main source of
dissolved Si is silicate mineral dissolution[37]. The negative
relationship between DOC and Si is explained by the dissolved
solids accumulated due to water-rock interaction in older
groundwaters[46]. In surface waters, lakes and streams with short
water residence times are biogeochemical hotspots where DOC is
rapidly produced and consumed[47]. These deeper and older
groundwaters are more likely to be depleted in DOC due to
oxidation processes, biodegradation and adsorption to soil and
aquifer mineral surfaces[48].

**Land use.** There is a significant increase of 19% in groundwater
DOC concentrations in urban areas compared to natural land.
Urban land use releases DOC to groundwater through leaking
sewage, landfill leaching, animal waste, fertilizer run-off, and
industrial and residential waste[27,49–51]. Within natural and agri-
cultural areas, there is a significant decrease in median groundwater
DOC concentrations where the subsoil clay fraction is > 30% ($n =$
2127) compared to subsoils with clay fraction ≤30% ($n =$ 2372,
Supplementary Fig. 7). In contrast, low (< 1%, $n =$ 4382) and high

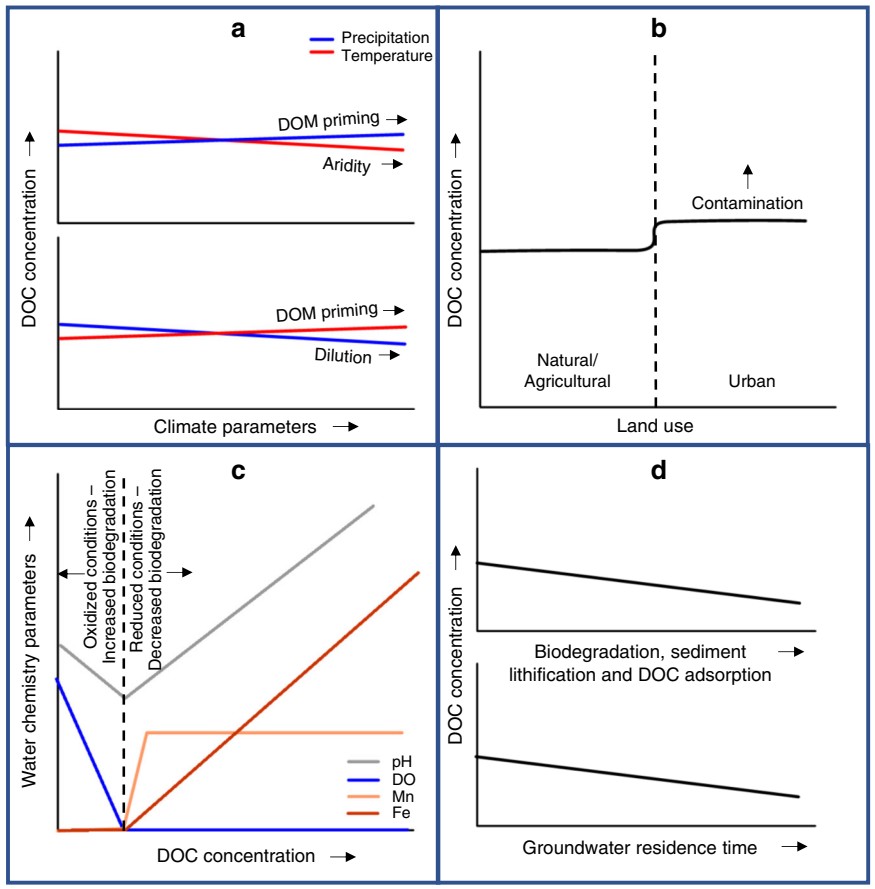

**Fig. 3 Conceptual model summarizing drivers of change in groundwater DOC concentrations. a** Climate parameters: in arid climates, groundwater DOC concentrations increase with increased precipitation due to the priming of organic matter by microbes under warm and increasingly wet conditions. Increased temperatures in arid environments reduced groundwater DOC due to increasing aridity. Precipitation in humid environments decreases groundwater DOC concentrations due to dilution, while temperatures increase DOM priming by microbes. **b** Urban land use contributes to groundwater DOC through contamination, for example through leaking septic and sewer systems. **c** Water quality parameters and groundwater DOC concentrations are linked and are largely controlled by redox conditions (NB: this panel shows variables, where DOC is the driver for the observed changes in water chemistry). **d** Aquifer age results in a decline in groundwater DOC due to sediment lithification and a depletion of sedimentary organic matter over time. Longer groundwater residence times decreased DOC by a combination of filtration of DOC through smaller aquifer pore sizes and adsorption (where residence times correspond to longer flow paths), and increased exposure to biodegradation over time.

(>1%, $n = 106$) soil organic carbon content within natural and agricultural areas do not appear to influence groundwater DOC concentrations ($p = 0.472$, Supplementary Fig. 7) suggesting that adsorption in interlamellar spaces or complexation with Fe[52] may play a more important role driving groundwater DOC than overlying soil organic carbon content. Our global dataset supports local scale observations showing that urban land use increases DOC in surface and groundwaters[53–55], showing that urban land use also impacts groundwater systems on a broader scale. We found no significant difference ($p = 0.841$) in groundwater DOC concentrations between natural and agricultural areas. Our finding of increased DOC in urban areas from a space-for-time analysis, as well as a previous space-for-time analysis[55], cannot reveal how this increase has occurred over time. A search for available groundwater TOC and DOC timeseries data in urban areas produced two datasets from Florida, US, and Perth, Australia. These data, collected from the 1980's to present in largely-residential urban areas, show no clear trend in groundwater DOC (Supplementary Fig. 8) over this timescale. Longer groundwater DOC timeseries on time scales longer than aquifer residence times would be needed to confirm our space-for-time interpretation. For example, fluvial DOC concentrations in the Thames Basin since 1883[56] have increased since World War 2 due to sewerage inputs and changes in

land management. Further groundwater DOC time series observations are required to assess the impact of urban area expansion, for example into lowland regions where DOC might be high, mobilization of previously stable soil DOC following development, and legacy contamination of groundwater in urban areas.

**Implications**. Continental-scale changes to groundwater DOC concentrations respond to changing temperature and precipitation patterns. Using our results and IPPC[5] (CMIP[5]) climate projections (www.worldclim.org), we identify more extreme groundwater DOC concentration changes associated with changing temperatures modulated by changing precipitation rates and patterns (Fig. 4). We identify hotspots of high groundwater DOC concentration (increases of up to 45%) associated largely with increased temperatures in the wettest quarter of the year in a number of south eastern US states under the business-as-usual Intergovernmental Panel on Climate Change (IPCC) climate change prediction scenario RCP8.5 (Fig. 4). Increasing temperatures stimulate phenol oxidase activity[57], which increases surface water DOC by 5.4% per year in the United Kingdom[58]. Importantly, relatively recalcitrant phenolic compounds[59] are selectively released as a result of this process. Therefore, under warmer

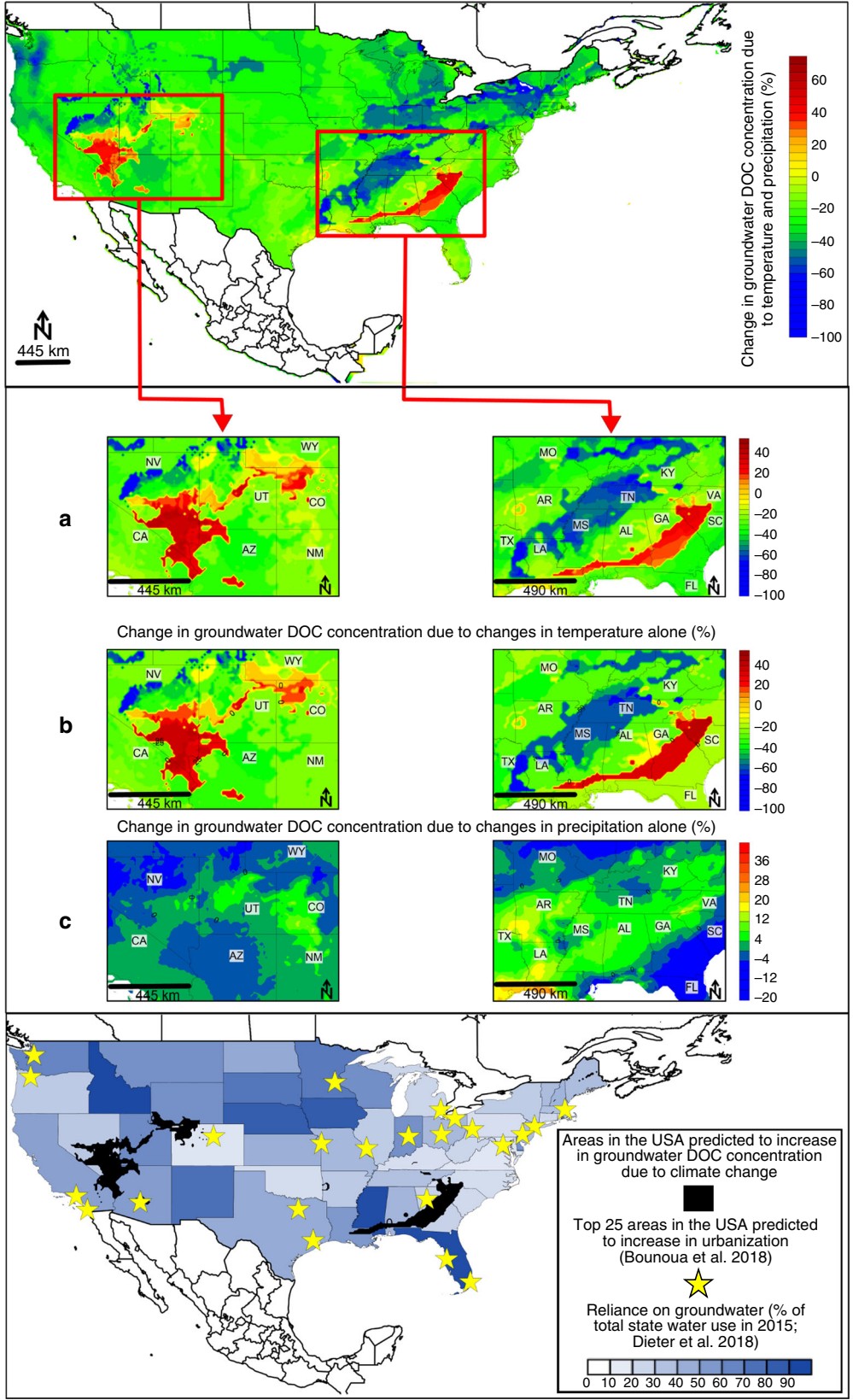

conditions, more DOC may persist along a flow path and ultimately enter groundwater systems. This increased carbon loading in groundwaters can change redox conditions and terminal electron acceptor availability for microbial use. This could drive

changes in groundwater microbial communities[60] and potentially enhance their survival rates.

The areas most at risk of future groundwater DOC concentration increases are those facing increased DOC due to climate change and

**Fig. 4 Changes in groundwater DOC concentrations by the year 2050 due to temperature and precipitation changes. a** Two areas in the US predicted to experience the largest increases in DOC concentration due to temperature and precipitation changes by 2050. **b** Changes in DOC concentrations in these areas caused by temperature variables (temperature in the wettest and warmest quarters of the year) alone. **c** Changes in DOC concentrations in these areas caused by precipitation variables (precipitation in the wettest and driest months of the year) alone. Groundwater DOC concentration changes are calculated using model results and IPPC5 (CMIP5) climate projections from the end of the 20th century (average of values from the period 1960 – 1990) to year 2050 (average of predicted values for the period 2041 – 2060)[30] for a "business-as-usual" climate change scenario (RCP8.5) as outlined in IPCC[71]. The lowermost map shows US state reliance on groundwater as a percentage of the total water use[72], overlain with areas predicted to experience increases in groundwater DOC concentrations due to climate change variables and urbanization.

where urbanization is predicted to occur. It is likely that DOC leaching will increase due to urbanization and population pressure on waste disposal networks. This will be a particularly significant issue for regions with a large or increasing reliance on groundwater as a source of fresh water. For example, areas in the US predicted to be impacted by DOC increases associated with climate have 13 to 87% reliance on groundwater as their source of fresh water (Fig. 4). Some of those regions were also projected to experience large increases in urbanization by 2020[61]. Our analysis suggest that this could lead to increased groundwater DOC concentrations, however these results are based on a space-for-time analysis. DOC time series in groundwater over time scales longer than aquifer residence times are required to confirm our model.

On a global scale, 54% of the worlds' population live in urban areas. By 2050 the world's urban population will increase to 66%[62]. By 2030, urban area is predicted to increase by 1.2 million km$^2$ [63]. This presents a significant issue when combined with the current lack of adequate sanitation services[64] maximizing the pollution of groundwater systems. Areas likely to see major urban expansion and population growth include eastern China, India and parts of Africa[63]. These areas are already facing high urban growth, have high population without basic sanitation levels (25 and 56% and up to 64% respectively)[64], and experience severe groundwater contamination issues that threaten local livelihoods[65,66]. Groundwater quality issues in south-eastern China may be further compounded by groundwater DOC increases associated with large predicted increases in temperature (up to 10 °C) in the wettest quarter of the year by 2050[30].

In some locations, increased groundwater DOC concentrations associated with climate change and urban land cover will increase the scale and hence capital and operational costs of water treatment facilities for groundwater DOC removal. Besides the direct implications of increased DOC, organic matter degradation in Holocene and Pleistocene aquifers in countries including Vietnam and Bangladesh have also been shown to correlate with Fe, NH$_4$, and As due to reductive dissolution of Fe-oxides in sediments where As is associated with Fe-oxides[3,67]. An increase in these dissolved species reduces groundwater quality and affects human health.

One common method of DOC removal from drinking water is via adsorption onto granular activated carbon. Increasing groundwater DOC concentrations in certain locations will require water treatment facilities to implement granular activated carbon as a second stage filter for DOC removal. For example, the current cost of water for a family of four is approximately US$845 per year in the US[68]. The implementation of granular activated carbon filtration methods by a 6.6 mega gallon per day facility (approximately 25 mega liter per day) would increase water costs for a family of four by US$134 per year (Supplementary Table 7, Supplementary Note 2). This equates to a 16% increase in annual household water costs in areas of Nevada, Georgia and South Carolina.

Overall, our investigation reveals that changes in climate and urban land cover are likely to impact groundwater DOC concentrations globally and regionally. These impacts on groundwater DOC will not be evenly distributed. Increases in temperatures in the warmest quarter of the year will decrease

groundwater DOC concentrations due to aridity, whilst warmer temperatures in the wettest quarter of the year will raise groundwater DOC by stimulating microbial activity. We identify hotspots of high groundwater DOC concentrations in areas that will undergo future urbanization and population growth. This could substantially increase the treatment costs to remove DOC from groundwater in many locations. Our results indicate that climate change and urban land cover will not only impact the quantity of the groundwater resource[23], but can also decrease groundwater quality and increase water treatment costs.

## Methods

**Literature survey and spatial coverage**. Google Scholar, Scopus as well as public data sources were searched using terms DOM, DOC, dissolved organic carbon, dissolved organic matter, groundwater quality for datasets presenting original (i.e., non-summarized) values of DOC. A number of authors, government departments and colleagues also provided original published and unpublished datasets. A total of 7849 unique groundwater DOC observations were obtained from published and unpublished datasets (Supplementary Table 1) after eliminating samples with a negative concentration value ($n = 36$), or those flagged with a V in the National Water Quality Assessment (NWQA) data ($n = 461$). The authors were advised that samples flagged with a V were suspected of being contaminated by methanol used in cleaning of the sampling apparatus and therefore they were excluded from the analysis. The data represents DOC concentrations for 31 countries on six continents with samples measured between 1992 and 2018. The authors excluded datasets reported to have been sampled from aquifers known to be heavily contaminated. A large proportion of the data come from samples obtained in US ($n = 5704$), followed by Australia ($n = 780$), Scotland ($n = 270$), England and Wales ($n = 113$), Zambia ($n = 110$) and Czech Republic ($n = 104$), with a lesser proportion of samples obtained from Malawi ($n = 89$), India ($n = 79$), Uganda ($n = 71$), Canada ($n = 52$), Ethiopia ($n = 44$), Nepal ($n = 40$), Poland ($n = 40$), Kenya ($n = 36$), Nigeria ($n = 35$), China ($n = 34$), Brazil ($n = 30$), Portugal ($n = 28$), Iceland ($n = 24$), Senegal ($n = 22$), Denmark ($n = 20$), Estonia ($n = 19$), Belgium ($n = 18$), Cook Island ($n = 17$), Argentina ($n = 15$), Bangladesh ($n = 13$), Mali ($n = 12$), Spain ($n = 10$), Malta ($n = 8$), France ($n = 7$), and Algeria ($n = 5$). We used these data to determine average DOC concentrations globally. 5459 samples from the US dataset which represent the data collected by the National Water Quality Assessment (NAWQA) program of the US Geological Survey (USGS)[29], were then used to generate the linear mixed model. This dataset was selected due to its large number of samples, ancillary data (inorganic water quality parameters) and availability of coordinates which allowed for the extraction of climatic, land use and unsaturated zone thickness data.

**Global groundwater DOC distributions**. DOC concentrations for each country were collected from the sources shown in Supplementary Table 1. Non-parametric unpaired one tailed Wilcoxon tests were used to identify differences in the groundwater DOC concentrations between continents, aquifer types and recharge rates using the function wilcox.test() in RStudio.

**Worldclim climate data**. High resolution (30 arc second) global ESRI grids were obtained for bioclimatic variables from www.worldclim.org (v1.4)[30]. Bioclimatic variables including annual mean temperature and precipitation, mean temperature and precipitation of the driest and wettest quarters, mean temperature and precipitation of the warmest and coldest quarters, as well as annual temperature range, isothermality, temperature and precipitation seasonality were extracted to each sample location using ArcGIS (v10.4.1). Where Worldclim data output showed that data was unavailable, as indicated by a −999.9 or −9999 value, these were removed and left blank.

**Land use data**. Land use data was obtained for the US dataset using a shapefile developed by the University of Maryland, Department of Geography and NASA[32,33]. Land uses were assigned to each sample location coordinate using ArcMAP (v10.4.1). Nineteen land use classifications are used in the file. Land use classifications were

reassigned to agricultural ($n = 3047$) where the land use type included the word cropland. Areas were assigned as urban ($n = 956$) for any area listed as urban/built up. Areas were assigned as wilderness ($n = 257$) for any samples containing the keywords forest, shrublands, wetlands, marsh, water and savannahs.

**Unsaturated zone thickness data**. Unsaturated zone thickness data[31] and was downloaded through GLOWASIS, the European Union collaborative project of Global Water Scarcity Information Service, at https://glowasis.deltares.nl/thredds/catalog/opendap/opendap/Equilibrium_Water_Table/catalog.html. Data was extracted to each sample location using ArcGIS (v10.4.1).

**Model development and statistics for US dataset**. A mixed linear model was developed using climatic data, land use data and unsaturated zone thickness data as well as parameters available in the US dataset available at https://doi.org/10.1594/PANGAEA.896953. Parameters included DOC, dissolved oxygen (DO), dissolved iron (Fe), sulfate ($SO_4$), magnesium (Mg), manganese (Mn), calcium (Ca), potassium (K), silica (Si), sodium (Na), fluoride (F), chloride (Cl), pH, sample temperature, sample depth below land surface, depth to the water table, land elevation, northing, precipitation in the wettest, driest, coldest and warmest quarters, maximum temperature in the warmest month, minimum temperature in the coldest month, temperature in the wettest, driest, coldest and warmest quarters of the year, mean diurnal temperature range, temperature seasonality, annual temperature range, annual precipitation, precipitation seasonality, annual average temperature, precipitation in the driest month, aquifer age, and land use type. Aquifer age was selected as a random effect, with all other variables applied as fixed effects which were selected using the manual Akaike Information Criteria (AIC) based backward selection using the drop1 () function in RStudio. This function allows for the identification of the variable with the lowest AICs so that they can be removed from the model.

**Quality assurance procedures**. Prior to data analysis, the data set was screened for <X values, which indicate a limit of detection in the analysis. Where these were identified, the value was replaced with a randomized value between 0 and X to ensure that bias associated with assigning these data points as either 0 or 1/2× is eliminated. A number of data points were flagged as potentially being contaminated with methanol and these samples were removed from the dataset.

Accuracy of sample coordinates were checked by adding an XY map of sample coordinates to a world map using Golden Software Surfer® (v 13.6.618). Any samples that were not located in the correct area as indicated by their ID label was investigated for typological error in the assigned coordinate and corrected ($n = 3$).

Prior to model generation, the response variable DOC was log transformed to normalize the data. Any sample with a missing value for one of the variables was removed using the na.omit function in R (v 3.3.1). This resulted in a final n of 2916 complete sample points used in the model. Predictor variables were then individually fitted to a simple linear model with DOC as the response variable to check assumptions. Standardized residuals vs. fitted value plots (Supplementary Fig. 9), Q–Q plots (Supplementary Fig. 10) and boxplots of residuals (Supplementary Fig. 11) were examined for each quantitative model variable to check that the assumption of constant variance and normality held true for residuals. Collinearity was checked through a regression matrix using R (v 3.3.1), which confirmed the presence of multicollinearity between some variables. The variance inflation factors (VIFs) for each variable were also checked. Typically, a variable is considered collinear with another variable when the VIF is greater than 10[69]. Some literature however recommends removing variables with VIFs greater than 4 or 5[70]. Some variables were known to covary, and thus were removed from the list of variables. These including Mn (due to covariance with Fe), EC (due to covariance with ions), temperature in the coldest and warmest months (covariance with temperatures in the coldest and warmest quarters), temperature and precipitation seasonality (due to covariance with temperature and precipitation range). All remaining variables with VIFs greater than 4 were then removed, with the variable with the highest VIF removed first before re-running the code each time until all remaining variables had VIFs < 10. The variables with VIFs > 10 were removed in the following order; annual temperature, temperature in the coldest quarter, precipitation in the driest quarter, annual precipitation, isothermality, precipitation in the wettest quarter, precipitation in the warmest quarter, precipitation in the coldest quarter, chloride, temperature in the driest quarter, northing, mean diurnal range. The results are reported as a percent change in DOC concentration, with standard error reported where possible (i.e., for continuous fixed effects variables).

**Change in groundwater DOC due to climate change in 2050**. Contours for changes in groundwater DOC concentration due to climate change in 2050 were developed by using current climate grid files from Worldclim v1.4[30] and future IPPC5 (CMIP5) climate projections (www.worldclim.org). Current climate values for climate variables used in the model (temperature in the wettest quarter of the year, temperature in the warmest quarter of the year, precipitation in the wettest month of the year and precipitation in the driest month of the year) were subtracted from future IPPC5 (CMIP5) climate projection values (www.worldclim.org) for the same variables for the year 2050 under a business-as-usual representative concentration pathway (RCP8.5) using Surfer® v.11.0.642. As DOC units in the model are log natural concentration values, the difference from zero for the

exponents of the intercepts for the four variables were multiplied by the difference in current and future values using a positive or negative sign at the front of the exponent depending upon whether the variable was positively or negatively correlated with DOC concentration. This provides a percent change in groundwater DOC concentration due to change in each climate variable predicted for 2050. The four grid files containing change in DOC concentration associated with each of the four variables were then added together into a single grid file to get a total change in DOC (%) associated with climate change in 2050.

## Data availability

The US dataset analyzed during the study is available on data repository PANGAEA (https://doi.pangaea.de/10.1594/PANGAEA.896953). Sources of published datasets used in the global groundwater DOC analysis are available in the Extended Data-Tables supporting document. Unpublished datasets that support the findings of the global groundwater DOC analysis will be provided by the corresponding author upon reasonable request and with permission of the owner(s) of the data. Code will be provided upon request by the authors.

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

## Acknowledgements

The authors would like to thank various ANSTO personnel such as Robert Chisari, Alan Williams, Barbara Gallagher, Kelly Farrawell, Chris Dimovski and Jennifer Van Holst for sample preparation and analysis for a portion of the Australian data. The authors thank Wuhui Duan from UNSW Sydney for his assistance in the collection and translation of data published in Chinese. We also thank graphic designer Anna Blacka from UNSW Sydney's Water Research Laboratory for her assistance with the preparation of Fig. 2, Professor Richard Kingsford at UNSW Sydney for his comments on the paper and Frank Chapelle from USGS for providing data and assistance with this study. Thanks to statisticians Peter Geelan-Small, Eve Slavich, and Nancy Briggs from UNSW Stats Central for their advice and guidance in the model development and interpretation. This work was supported by the Australian Research Council [DP160101379], the Australian Government Research Training Program Scholarship and a scholarship provided by ANSTO. A large portion of the Australian DOC data was also made possible by grants from the Federal Government initiative National Collaborative Research Infrastructure Strategy (NCRIS), the Cotton Catchment Community (CRC-CCC), the NSW Department of

Primary Industries Office of Water and support for students via the National Centre for Groundwater Research and Training (NCGRT). D.J.L. and D.C.G. were funded through the LOCATE NERC grant NE/N018087/1. Ethiopia data used in the global dataset comparison was funded by Hidden Crisis 2 NE/M008606/1. Iceland data was funded through BGS-NERC Earth Hazards and Systems Directorate. Kenya data collection was funded through funded through HyCRISTAL NE/M020452/1. Mali and Nigeria data collection were funded through UK Department for International Development, GA/09 F/094, Groundwater resilience to climate change in Africa. Malawi data collection was funded through TRIGR, REACH programme and Hidden Crisis 2 NE/M008606/1. Nepal data collection was funded through UK Department for International Development, Groundwater Resources in the Indo-Gangetic Basin, Grant 202125-108. Senegal data collection was funded through AfriWatSan, Royal Society DFID Africa Capacity Building Initiative, ref: AQ140023. Uganda data collection was funded through HyCRISTAL NE/M020452/1 and Hidden Crisis 2 NE/M008606/1. Brazil, Argentina and a number of Australian data used in the global analysis were collected with funding from the Australian Research Council (FT170100327).

## Author contributions

L.K.M. drafted the main paper and was responsible for model development and figure preparation. A.B., M.S.A., D.M.O., H.R., and K.M. supervised the project and provided paper comments and guidance. I.R.S. provided unpublished data, guidance on the direction of the paper and paper comments. P.O. assisted with sourcing of data and provided advice on DOC sources and sinks. J.B. provided water treatment costs associated with groundwater DOC removal and provided calculations on increases in household water costs associated with future temperature and precipitation changes. D.C.G, J.P.R.S, D.J.L., A.M.M., and J.W. provided published and unpublished groundwater datasets and paper comments.

## Competing interests

The authors declare no competing interests.
