## [Peer Review File · Nature Communications]

Reviewers' comments:

Reviewer #1 (Remarks to the Author):

I agree with Reviewer 2's statements regarding the appropriateness of the article content for a broad readership journal (e.g., "However, I don't feel the authors haven't really made the case for the major significance of the results that would warrant publication in Nature Geosciences"). We each made similar points in our earlier reviews; this work would fit better in a more field-specific journal.

I thank the authors for their replies to the first round of reviews. A remaining technical concern lies in the authors' reply to my fourth bullet point ("4. Figure 1 – country-wide medians are perhaps a bit misleading given the substantial spatial variations that can only be impacted indirectly by national borders. Perhaps a global aquifer map (e.g., WHYMAP) could fit better?"). I recommend revisions to address this comment, beyond the addition of box plots referred to in the reply letter.

Reviewer #2 (Remarks to the Author):

This paper asserts that future climate change and urbanization will be the major drivers of groundwater quality. This is considered, using dissolved organic carbon (DOC) as a proxy for groundwater quality. The work is well-written and well-presented with clear figures and detailed descriptions of the various datasets used, techniques applied, and a discussion of the potential interpretation of the results.

There are clear links to recent work on groundwater at a global scale, and the datasets used provide an excellent facility to extend the consideration of global groundwater resource issues to the matter of water quality. At such a scale it is perhaps reasonable to use statistical methods to identify key linkages between drivers and responses, but in order for this to be established as a viable hypothesis there needs to be a mechanistic model that provides a more substantive process-understanding to underpin the resulting discussion.

I don't think the manuscript is suitable for publication in its present form because it does not establish: (1) Clear, mechanistic causal links between potential drivers and groundwater DOC concentrations. The model used is a statistical regression tool to generate associations between some potential drivers and responses and these are then presented as causal links, but without introducing a mechanistic model to explain these; (2) How different recharge mechanisms would affect the results, and how the time delays associated with differential recharge rates would impact on the use of groundwater quality samples as being representative of other driving variables at that time; and, (3) Why the major drivers of DOC (at a river-basin scale) are not considered, i.e. the presence and extent of organic and organo-mineral soils.

Further, the manuscript quickly starts to refer to groundwater DOC as a more general marker of groundwater quality, although there isn't a strong justification as to why this should be taken as a global indicator in the context of other geochemical indicators. The question as to what concentration of groundwater DOC is problematic (for potable treatment or as a benchmark of quality) is also more complex than presented. Much of the attention on trihalomethane production (from the water treatment process) stems from the aesthetic standard for water colour (which is not harmful, just undesirable), and the treatment of potable waters to remedy water colour requires the removal of the DOC (which is the main cause of discolouration). However, without a thorough discussion of limits of dissolved C for potable water, there is no strong argument as to why this is a pressing issue and, therefore, why groundwater DOC should be a focus.

I can see that there is some compelling evidence that climate and urbanization may be useful proxies for understanding variability in observed groundwater DOC – the statistical model certainly shows that

– but the datasets for which these results emerge are not really global. This is perhaps inevitable, but there is a large extrapolation between the statistical results and the global scale that is rather difficult to make in isolation from other potential influences on global groundwater quality.

For these reasons I do not think that the manuscript is suitable for publication at this time. There is clearly an interesting narrative around groundwater DOC, but without significant additional work, I do not see how the conclusions proposed here can be reached.

Reviewer #3 (Remarks to the Author):

Both reviewers questioned whether this paper presents sufficiently novel result for a Nature publication. Reviewer #1 suggests that, “The ramifications of the results are of lesser impact than other recent groundwater quality research published by some of the coauthors.” Reviewer #2 states that “It just doesn’t communicate a message that is of the level of significance for the scientific community that is normally expected from a paper in Nature Geoscience”. I also did not learn anything transformative from this paper, so I think the reviewers identified a major concern.

However, I think this manuscript should be published in Nature Communications both because it is a scientifically-solid and useful addition to the literature, bringing a variety of insights, mostly already known, together in one manuscript and also because this manuscript stands-out as high quality relative to other global assessments of water resources that have been published in Nature journals. It seems unfair not to publish such a clearly written and thoughtful assessment after Nature has published global assessments of water resources with much less in the way of new scientific insights.

I found that the reviewers’ other comments and concerns were addressed by the authors’ edits and responses. I also have several other suggestions to strengthen the paper.

- Can the authors find any DOC time series data that show groundwater DOC concentrations changing, or not, over time with urbanization? It seems such time series should be available from urban groundwater supplies. Observations of change with urbanization would greatly strengthen their argument that is now based only differences across locations. It’s good to temporal data to argue for a temporal trend.
- I think the authors should caution readers from interpreting the distributions shown in figure 1, top panel, as demonstrating differences among countries. With the possible exception of the large data sets from the US and Australia, I suspect that the differences in DOC concentrations more likely reflect differences in local sampling choices than difference across countries.

Charles Harvey

Changes in global groundwater organic carbon driven by climate change and urbanization

Response to reviewer comments

Reviewer Comments:

Reviewer #1 (Remarks to the Author):

I agree with Reviewer 2's statements regarding the appropriateness of the article content for a broad readership journal (e.g., "However, I don't feel the authors haven't really made the case for the major significance of the results that would warrant publication in Nature Geosciences"). We each made similar points in our earlier reviews; this work would fit better in a more field-specific journal.

We believe that the manuscript is a valuable contribution to the broader scientific readership and has wide implications. We reveal the mechanisms controlling groundwater DOC using a large and statistically powerful dataset. Whilst DOC has been discussed in the literature, this has typically focused on one or a limited number of sites. As such the relationships between DOC and various parameters, as well as their significance on a global scale, have not been investigated. Our model relies on a large dataset to determine widespread relationships between DOC and numerous parameters (land use, climate, chemistry and aquifer age / groundwater evolution) of interest for multiple disciplines (geography, climatology, biogeochemistry, hydrology and environmental engineering).

We go well beyond several recent groundwater papers published in recent Nature journals (as also noted by Reviewer 3), giving us confidence the topic belongs to a journal in the Nature family.

We have further strengthened the paper by adding a mechanistic conceptual model, and added further information on public health and water quality implications (see further detail in the response to Reviewer #2's comments). We have identified some additional DOC datasets increasing our sample size from 7,849 to 8,010 which have been added to Table S1 and the analysis. We have also performed additional analysis regarding the impact of aquifer properties and soil parameters using WHYMAP and Regridded Harmonized World Soil Database data (aquifer type, aquifer recharge rates and subsoil clay and OM content), and identified that aquifer types, recharge rates and subsoil clay content impact significantly on groundwater DOC. These are discussed further in response to the comment below as well as in the response to Reviewer #2's comments. We have also investigated time-series properties to further help elucidate the processes occurring in urban areas (see response to Reviewer #3's comments for further details).

I thank the authors for their replies to the first round of reviews. A remaining technical concern lies in the authors' reply to my fourth bullet point ("4. Figure 1 – country-wide medians are perhaps a bit misleading given the substantial spatial variations that can only be impacted indirectly by national borders. Perhaps a global aquifer map (e.g., WHYMAP) could fit better?"). I recommend revisions to address this comment, beyond the addition of box plots referred to in the reply letter.

We agree that DOC concentrations do not necessarily recognize country or administrative borders. We have therefore replaced the continent comparisons with aquifer data from WHYMAP as recommended, including new Figures S1 and S2. We note that latitude and longitude data are largely only available for the USGS dataset. We therefore extracted aquifer type and recharge rate information and provide a further assessment on how these aquifer variables affect groundwater DOC concentrations in the U.S. The following text has been included:

Line 86 – 95: *"Variations in DOC concentrations between countries (Fig. 1) are likely to be related to recharge rates and aquifer types. World-wide Hydrological Mapping and*

Assessment Programme (WHYMAP) data ²⁸ suggests that within the U.S. dataset (Fig. S1 ²⁹), major groundwater basins contain significantly lower DOC concentrations than local and shallow aquifers, and complex hydrogeological structures (both $p < 2.2 \times 10^{-16}$, Fig. S2). Therefore, groundwater age and depth seem to control groundwater DOC. There are also significantly higher DOC concentrations identified in aquifers with < 100 mm / year recharge compared to those with high recharge rates (100 – 300 mm / year, $p = 2.342 \times 10^{-7}$) and very high recharge rates (> 300 mm / year, $p = 4.857 \times 10^{-5}$, Fig. S2), which could indicate a dilution effect.”

We have also included a map of the USGS dataset overlain over a WHYMAP global aquifer map in supplementary Fig. S1, and a boxplot comparison of aquifer types and recharge rates for the USGS data in Fig. S2:

Fig. S1.

U.S. NWQMC data (red plus symbols) ¹ overlain over global aquifer WHYMAP data ².

Fig. S2.

Comparison of groundwater DOC concentrations between aquifer type and recharge rates for U.S NWQMC data using global aquifer WHYMAP data ². Median DOC concentrations in major groundwater basins were significantly lower than in complex hydrogeological structures or local and shallow aquifers (both $p < 2.2 \times 10^{-16}$). Groundwater DOC concentrations in aquifers with medium – low recharge rates (< 100 mm / yr) were significantly higher than in aquifers with 100 - 300 mm / year and > 300 mm / year ($p = 2.342 \times 10^{-7}$ and 4.857×10^{-5} respectively). Outliers greater than 6 mg / L have been removed for clarity ($n = 154$ [complex hydrogeological structures], $n = 26$ [local and shallow aquifers], $n = 107$ [major groundwater basins], $n = 15$ [very high aquifer recharge (> 300 mm / yr)], $n = 153$ [high aquifer recharge (100-300 mm / yr)] and $n = 119$ [medium – low aquifer recharge (< 100 mm / yr)]).

We thank reviewer 1 for their comments.

Reviewer #2 (Remarks to the Author):

This paper asserts that future climate change and urbanization will be the major drivers of groundwater quality. This is considered, using dissolved organic carbon (DOC) as a proxy for groundwater quality. The work is well-written and well-presented with clear figures and detailed descriptions of the various datasets used, techniques applied, and a discussion of the potential interpretation of the results.

We appreciate these positive comments.

There are clear links to recent work on groundwater at a global scale, and the datasets used provide an excellent facility to extend the consideration of global groundwater resource issues to the matter of water quality. At such a scale it is perhaps reasonable to use statistical methods to identify key linkages between drivers and responses, but in order for this to be established as a viable hypothesis there needs to be a mechanistic model that provides a more substantive process-understanding to underpin the resulting discussion.

We've now added a new figure (Fig. 3) which shows a mechanistic process-based understanding for each of the four categories of variables identified as being significant in the model (i.e. climate, land use, water chemistry and aquifer / groundwater age):

Line 149 – 164:

“Fig. 3. Conceptual model showing the mechanisms for change in groundwater DOC concentrations. (A) Climate parameters: in arid climates, groundwater DOC concentrations increase with increased precipitation due to the priming of organic matter by microbes under warm and increasingly wet conditions. Increased temperatures in arid environments lead to decreased groundwater DOC due to increasing aridity. Increased precipitation in humid environments decreases groundwater DOC concentrations due to dilution whilst increased temperatures increase DOM priming by microbes. (B) Urban land use contributes to groundwater DOC through contamination, for example through leaking septic and sewer systems. (C) Water quality parameters and groundwater DOC concentrations are linked and are largely controlled by redox conditions (NB: C shows variables where DOC is the driver for the observed changes in water chemistry). (D) increasing aquifer age result in a decline in groundwater DOC due to sediment lithification and a depletion of sedimentary organic matter over time. Increasing groundwater residence times lead to decreasing DOC by a combination of filtration of DOC through smaller aquifer pore sizes and adsorption (where residence times correspond to longer flow paths), and increased exposure to biodegradation over time.”

We feel that this figure helps to strengthen the discussion by providing a conceptual mechanistic model of the processes resulting in the statistically significant model correlations that we observe, and we thank the reviewer for this suggestion.

I don't think the manuscript is suitable for publication in its present form because it does not establish: (1) Clear, mechanistic causal links between potential drivers and groundwater DOC concentrations. The model used is a statistical regression tool to generate associations between some potential drivers and responses and these are then presented as causal links, but without introducing a mechanistic model to explain these;

Please refer to the above for our response regarding mechanistic, casual links between potential drivers and groundwater DOC concentrations.

(2) How different recharge mechanisms would affect the results, and how the time delays associated with differential recharge rates would impact on the use of groundwater quality samples as being representative of other driving variables at that time; and, (3) Why the major drivers of DOC (at a river-basin scale) are not considered, i.e. the presence and extent of organic and organo-mineral soils.

As this model is based on a very large dataset, the relationships between DOC and each of the significant parameters represent the dominant relationships present in the dataset. Smaller scale studies may reveal different relationships than the ones captured by our large scale model (for example, our discussion on the Australian dataset in the supplementary information). We have added a comment on this in the manuscript:

Line 131 – 133: *“While the model represents large scale relationships between DOC and control variables, these relationships can vary on a site by site basis due to site specific variables.”*

We feel that factors such as recharge mechanisms and time-lags would largely be controlled by site or sample specific variables (e.g. distance to surface waters, hydraulic conductivity etc.) which are not available for individual DOC samples. For this reason, we cannot make conclusions regarding how recharge rates and time-lags may be represented by, or interact with the other model variables. Further, we feel that the suggested analysis would be more suited to local or river-basin scale investigations. This paper combines multiple local scale datasets, incorporating samples from thousands of sites including various river basins and varying recharge mechanisms, rates and time-lags. We use this data to produce the first global scale analysis of mechanisms controlling DOC. A focus on local or river-basin scale drivers would prevent us from demonstrating the link between global climate change and groundwater chemistry. We consider this link a major finding of the manuscript with important implications.

We have however included new discussion on how subsurface clay content and organic carbon content may impact on groundwater DOC. We extracted subsoil clay and organic matter content data for each datapoint in the USGS database from the RegridDED Harmonized World Soil Database and have added the following to the manuscript:

Line 251– 258: *“Within natural and agricultural areas there is a significant decrease in median groundwater DOC concentrations where the subsoil clay fraction is > 30% (n = 2,127) compared to subsoils with clay fraction ≤ 30% (n = 2,372, Fig. S9). In contrast, low (< 1%, n = 4,382) and high (> 1%, n = 106) soil organic carbon content within natural and agricultural areas do not appear to influence groundwater DOC concentrations (p = 0.4723, Fig. S10) suggesting that adsorption in interlamellar spaces or complexation with Fe⁵³ may play a more important role in determining groundwater DOC concentrations than overlying soil organic carbon content.”*

We have also added the following plot to the supplementary information:

Fig. S9.

Comparison of groundwater DOC concentrations with varying subsurface clay and organic matter content using RegridDED Harmonized World Soil Database v1.2³. Median DOC concentrations in soils with high clay percent weight (> 30%) are significantly lower than areas where the subsoil clay percent weight is medium (10 – 30%, $p = 5.628 \times 10^{-13}$) or low (< 10%, $p = 5.533 \times 10^{-8}$). Groundwater DOC concentrations in aquifers with low (0 – 1%) and high (> 1%) soil organic carbon content are not significantly different ($p = 0.4723$). Outliers greater than 5 mg / L have been removed for clarity ($n = 25$ [low (< 10%) clay fraction in subsoil], $n = 108$ [medium (10 – 30 %) clay fraction in subsoil], $n = 124$ [high (> 30%) clay fraction in subsoil], $n = 253$ [low (0 – 1%) organic carbon content in subsoil], and $n = 4$ [high (> 1%) organic carbon content in subsoil]. N.B. this data represents agricultural and natural areas only due to the potential for paved urban areas to affect infiltration of DOC through the subsoil.

Further, the manuscript quickly starts to refer to groundwater DOC as a more general marker of groundwater quality, although there isn't a strong justification as to why this should be taken as a global indicator in the context of other geochemical indicators. The question as to what concentration of groundwater DOC is problematic (for potable treatment or as a benchmark of quality) is also more complex than presented. Much of the attention on trihalomethane production (from the water treatment process) stems from the aesthetic standard for water colour (which is not harmful, just undesirable), and the treatment of potable waters to remedy water colour requires the removal of the DOC (which is the main cause of discolouration). However, without a thorough discussion of limits of dissolved C for potable water, there is no strong argument as to why this is a pressing issue and, therefore, why groundwater DOC should be a focus.

We have strengthened the justification as to why groundwater DOC is a global indicator of groundwater quality. This includes recent research by Evans et al. (2019, DOI: 10.1016/j.scitotenv.2005.01.027) who identified that over 100,000 lifetime cancer cases in the U.S. may be due to arsenic, disinfection by products and radioactive contaminants in drinking water. Both arsenic and DBPs are strongly linked to DOC concentrations. This has been included in the introductory text. We have also added a thorough discussion of the limits of DOC for potable water in the manuscript (these are typically very low at > 5 mg / L), as well a reference to the low

concentrations of DOC which can impact on water treatment costs due to membrane fouling to provide more context as to why groundwater DOC is a pressing issue and should be a focus:

Line 36 – 57: *“Over 100,000 lifetime cancer cases in the United States (U.S.) can be attributed to contaminants in drinking water. A large proportion of the risk identified is associated with the presence of disinfection by products (DBPs) and arsenic⁷, both of which are strongly linked to DOC^{3, 8, 9, 10, 11}. Chlorination and ozonation used for water treatment can result in harmful by-products including 3-chloro-4-dichloromethyl-5-hydroxy-2(5H)-furanone, brominated acetic acid, trihalomethanes (THM), formaldehyde, halogenated acetic acids, due to the presence of organic matter¹². These by-products can be genotoxic, carcinogenic or result in tumors¹². Since most of the health impacts caused by DOM are related to the formation of by-products and depend on the concentrations of other water chemistry parameters, the World Health Organisation¹² and many countries including Australia¹³ do not regulate total organic carbon (TOC) or DOC concentrations in drinking water directly but many countries such as USA¹⁴, Canada¹⁵, France¹⁶, China¹⁷ and South Africa¹⁸ highlight potential concerns related to THM formation, health effects and aesthetic quality in the broad DOC range of 0 - 5 mg / L during treatment.*

In addition to health and aesthetic impacts, the presence of DOC in water can lead to membrane fouling after ozonation. In order to avoid this, a biological filtration step is advised to be added to the water treatment process for water containing DOC concentrations > 1 mg / L¹⁹. This indicates that even relatively small DOC increases in raw groundwaters can have impacts not only on human health and water aesthetics, but also on the ease and cost of water treatment.”

I can see that there is some compelling evidence that climate and urbanization may be useful proxies for understanding variability in observed groundwater DOC – the statistical model certainly shows that – but the datasets for which these results emerge are not really global. This is perhaps inevitable, but there is a large extrapolation between the statistical results and the global scale that is rather difficult to make in isolation from other potential influences on global groundwater quality.

We agree that the statistical model provides some compelling evidence for climate and urbanisation controls on groundwater DOC. As indicated by Reviewer 1, DOC concentrations are not controlled by country borders, so whilst the dataset is based on data collected in the continental U.S., the samples represent a wide range of environments that are present on the globe, and represent data from 17 aquifer age groups ranging from Cambrian age to Quaternary age. We have added a new Table S7 to the supplementary information which shows the annual average temperature and precipitation ranges in the

Table S7.

Summary table of annual average temperature and precipitation in the U.S. NWQMC dataset.

	Annual average temperature	Annual average precipitation (mm)
Min.	2.8	94.0
1st Qu.	9.2	480.2
Median	11.5	812.0
Mean	12.6	809.1
3rd Qu.	16.5	1133.0
Max.	24.2	1798.0

The process-based understanding derived from this continental scale analysis has therefore allowed us to develop the mechanistic model for groundwater DOC that is likely to be applicable for most environments.

For these reasons I do not think that the manuscript is suitable for publication at this time. There is clearly an interesting narrative around groundwater DOC, but without significant additional work, I do not see how the conclusions proposed here can be reached.

We have now performed a significant amount of additional work to address the comments provided by Reviewer 2. This includes:

- The addition of new groundwater DOC datasets;
- Undertaking additional literature review to convey the importance of investigating DOC. This includes examining the DOC concentration limits in drinking water guidelines from numerous countries around the world, adding reference to recent research showing that disinfectant by products and arsenic, both of which are strongly linked to DOC concentrations, cause over 100,000 lifetime cancer cases in the U.S., and adding a new reference which shows that small increases in DOC can contribute to membrane fouling in the distribution system after ozonation, increasing water treatment costs;
- Addition of a new Fig. 3 to show the mechanisms controlling groundwater DOC concentrations which supports the conclusions made in the text; and
- The extraction of soil property data including clay and organic matter content from the Regrided Harmonized World Soil Database for the U.S. datapoints, statistical analysis of how these properties impact on groundwater DOC, and new associated supplementary figures.

Reviewer #3 (Remarks to the Author):

Both reviewers questioned whether this paper presents sufficiently novel result for a Nature publication. Reviewer #1 suggests that, "The ramifications of the results are of lesser impact than other recent groundwater quality research published by some of the coauthors." Reviewer #2 states that "It just doesn't communicate a message that is of the level of significance for the scientific community that is normally expected from a paper in Nature Geoscience". I also did not learn anything transformative from this paper, so I think the reviewers identified a major concern.

However, I think this manuscript should be published in Nature Communications both because it is a scientifically-solid and useful addition to the literature, bringing a variety of insights, mostly already known, together in one manuscript and also because this manuscript stands-out as high quality relative to other global assessments of water resources that have been published in Nature journals. It seems unfair not to publish such a clearly written and thoughtful assessment after Nature has published global assessments of water resources with much less in the way of new scientific insights.

I found that the reviewers' other comments and concerns were addressed by the authors' edits and responses. I also have several other suggestions to strengthen the paper.

- *Can the authors find any DOC time series data that show groundwater DOC concentrations changing, or not, over time with urbanization? It seems such time series should be available from urban groundwater supplies. Observations of change with urbanization would greatly strengthen*

their argument that is now based only differences across locations. It's good to temporal data to argue for a temporal trend.

To assess whether time is a factor in increasing DOC in urban areas, we applied a sampling date parameter to determine whether DOC concentrations have been increasing over time for both the total dataset and for the urban subset of data. In both cases the drop1 function used for model selection removed the time parameter, suggesting that time is not significantly correlated with changes in groundwater DOC concentrations including in urban environments. We attempted to investigate this further by contacting water utilities and government departments to obtain timeseries data in urban areas. Contact was made with Denver Water (Colorado) who indicated that they do not have DOC data for groundwater available. Mount Pleasant Waterworks in South Carolina and Hunter Water (NSW, Australia) were also contacted however we did not receive a response from these companies. Hunter Water have publicly available drinking water quality summaries available on their website, and these appear to contain results for the water quality parameters which are regulated in Australia. This does not include DOC or TOC concentrations. We also attempted to contact a number of government departments. The Minnesota Department of Natural Resources Groundwater Unit informed us that they do not collect groundwater DOC. We did however receive data from Water Corporation in Perth, Australia, and from the Data Collection Bureau at South West Florida Water Management District who provided datasets for a number of sites located in a developing area of Perth, and on the south west coast of Florida respectively. These datasets showed a mix of trends including positive, negative or no trend which further confirmed that lack of relationship between DOC concentration and time in urban areas. This leads us to conclude that the higher median and mean DOC concentrations in urban areas (shown in the table below) may represent an increase in DOC potentially due to point source events or leaks which persist over time periods and which aren't captured in time series.

Land use	N	Mean	sd	median	min	max	range
Agricultural	3047	2.03	9.98	1.00	0.00	380.00	380.00
Natural	1451	2.67	28.49	0.56	0.00	1041.56	1041.56
Urban	956	2.92	8.59	1.13	0.00	223.22	223.21

Where we present or discuss the model results in the manuscript, we have removed references to “urbanization” and replaced it with “urban land cover” so as not to imply that the model shows a change over time.

• I think the authors should caution readers from interpreting the distributions shown in figure 1, top panel, as demonstrating differences among countries. With the possible exception of the large data sets from the US and Australia, I suspect that the differences in DOC concentrations more likely reflect differences in local sampling choices than difference across countries.

This is a very good point and we have added a line in the caption of Fig. 1 to address this:

Line 101 – 102: *“It is noted that the distributions shown in A may be influenced by local sampling bias, particularly for small datasets.”*

We thank all of the reviewers for their comments.

END OF REVIEWS

Reviewers' comments:

Reviewer #1 (Remarks to the Author):

Fig. S4's caption refers to 'strong' correlations, yet I suspect few would interpret scatter plot against groundwater temperature as strong.

A regression line should be added to each Fig. S4 subplot. Regression coefficients and their uncertainties should be clearly stated for each subplot of Fig. S4 in the caption (or in text within or above each subplot).

The national-scale box plots in Fig. 1 are strange. Groundwater does not recognize such national boundaries (though can be impacted indirectly by behavioral differences across borders). This comment was made previously. There exists important spatial variations in groundwater quality within nations that are not transparently presented in the existing figures.

Reviewer #2 (Remarks to the Author):

I think the authors should be congratulated on an excellent set of well-argued responses, thoughtful revisions, and a final manuscript that I think is an excellent contribution. I was not entirely convinced by the earlier drafts of this manuscript, but I feel the latest additions are a significant improvement. Thank you for taking the time to address the comments in such a thorough way.

I think the paper should be published in its current form.

Reviewer #3 (Remarks to the Author):

Second Review

I focus my review on the new figure added to this version of the paper and to the two suggestions I made in my previous review. In my opinion, all three of these issues must be addressed.

New Figure. I agree that the new figure helps to convey the conclusions of the paper. However, this figure greatly exaggerates the magnitude of the results. For example, panel B shows that DOC concentrations are double in urban than in natural areas. I added a red line showing the actual finding of the paper, a 19% difference. The red line accurately represents the finding of the paper while the original black line exaggerates the finding. It's important to understand that the difference between natural and urban locations found in this paper was a modest 19% to place the result in context of the variability of DOC concentrations which is often orders of magnitude. Some of the other panels also appear strongly exaggerate the findings: the lines representing DOC should be much closer to horizontal in panel A. Yes, the p-values that the statistical significance of the findings is strong. However, the effects of climate are modest whereas the figure implies, incorrectly, that DOC concentrations go to zero at the origin or on the right side of the graph.

I agree with the authors that this figure is useful because it is important to visually convey the direction of the effects, but is equally important to convey the magnitude of the effects. By adding values to the X-axes and adjusting the lines to represent the actual results, this figure can convey the findings, including the important finding that many of the effects are quite modest. I understand that this figure is meant as a “cartoon,” not as quantitative results. None-the-less, this figure will mislead readers as it is.

Previous Comment #1. This paper concludes that urbanization causes DOC concentrations to rise. This is why analysis of DOC data before and after urbanization is important. However, the result rests entirely on a “space for time” analysis, comparing concentrations across urban and natural sites. The problem with this analysis is that it cannot prove that DOC in natural areas will rise if those areas are urbanized. There could be other explanations for the observed 19% difference. For example, it could easily be that urban development has tended to be in low-lying areas near bodies of water where there are more wetlands that are a source of DOC. This seems a very plausible explanation for a 19% difference to me.

I appreciate that the authors worked hard to find time series of DOC data. In their response to my comment they, they describe how their analysis of data from Perth and Florida showed “a mix of trends including positive, negative or no trend which further confirmed the that lack of relationship between DOC concentration and time urban areas.” Why did they not include this in their paper?

The paper needs to give an objective analysis of the question of causation versus correlation. The authors argue that they do not have enough time-series data to answer the question, and chose not to include the only data they could find. As the paper reads now, many readers will think that the authors really did determine that DOC concentrations increase when land is urbanized. To avoid misleading readers, the paper must state up-front and prominently that they do not analyze any data describing how DOC concentrations change before and after urbanization. Rather, they perform a space-for-time analysis and the results of this are consistent with such a change, but are also consistent with other explanations. The result that DOC is higher in urban groundwater is useful, but should not be confused with the idea that urbanization causes DOC to increase.

Previous Comment #2. The authors need to go further in addressing this comment. To explain, I'll use the cases of Senegal and Nigeria. Senegal is shown with the highest DOC concentrations and Nigeria is shown with the eighth from the lowest. DOC appears to be about seven times higher in Senegal than Nigeria and the standard deviations for the two data sets do not overlap. Of course, to someone who has experience in this field, it is immediately obvious that this data cannot be used determine that Senegal has higher DOC concentrations because there was no effort to design a sampling scheme to find a national average in either country and their a very few data points in both countries. **But to other readers this will not be obvious and there can be real consequences.** For example, it is conceivable that an NGO could read this paper and conclude that a water treatment program should all focused in Senegal without any concern for Nigeria.

I would recommend replacing these box-and-whisker plots with different categories. Instead of countries, show categories for which the authors believe there is a real difference in DOC, such as age of aquifer or depth. If the authors insist on keeping the current plots, that differentiate the data by country, then they should clearly and prominently state that this plot cannot be used to determine country by country levels of contamination — even though that's what it appears to show.

Reviewer Comments:

Reviewer #1 (Remarks to the Author):

Fig. S4's caption refers to 'strong' correlations, yet I suspect few would interpret scatter plot against groundwater temperature as strong.

A regression line should be added to each Fig. S4 subplot. Regression coefficients and their uncertainties should be clearly stated for each subplot of Fig. S4 in the caption (or in text within or above each subplot).

Fig. S4 and Fig. S5 show scatterplots of DOC (log scale) versus control variables with positive and negative correlations representing significance levels of $P < 0.001$ in the model. These plots only represent the correlation between the two variables and do not account for the other variables included in the model. We have therefore clarified this in both captions for Fig. S4 and Fig. S5 (now Fig S.10 and S.11) to state "NB: these plots show only the correlation between $\log(\text{DOC concentration}(\text{mg} / \text{L}))$ and individual variables. They do not account for the other variables included in the model and do not represent model results".

To address the reviewer's request for the model regression coefficients and uncertainties, we have added a new figure (now Fig. S3) which summarizes the model regression coefficients and uncertainties (50% and 95% confidence intervals) for each variable:

Fig. S3.

Regression estimates of the effects of model variables on groundwater DOC concentrations. Centre points represent mean regression estimates with inner (thicker) bars representing 50% confidence intervals and outer (thinner) bars representing 95% confidence intervals. Regression estimates from top to bottom are 0.17, -9.74×10^{-3} , -6.29×10^{-3} , -5.66×10^{-4} , 3.75×10^{-5} , 2.83×10^{-2} , 4.45×10^{-3} , -9.53×10^{-3} , -7.77×10^{-3} , -9.61×10^{-2} , 4.49×10^{-2} , -7.06×10^{-2} , 6.87×10^{-5} , -2.45×10^{-3} , -9.33×10^{-2} , 3.35×10^{-2} , -2.66×10^{-4} and 1.84×10^{-3} (also listed in Table S.2).

The national-scale box plots in Fig. 1 are strange. Groundwater does not recognize such national boundaries (though can be impacted indirectly by behavioral differences across borders). This comment was made previously. There exists important spatial variations in groundwater quality within nations that are not transparently presented in the existing figures.

We agree groundwater does not recognize national borders and have changed Fig. 1 to show a histogram with mean and median of the aggregate results of DOC concentrations, with countries represented in the data shown in red in the lower (B) panel. We feel that this will avoid any misinterpretation of the data when data was separated into countries.

Fig. 1. A) histogram showing median global groundwater DOC concentrations (mg C L^{-1}). Sample sizes for individual countries ranged from 5 to 5,812, with 14 out of 32 countries having $n < 30$. We have therefore presented aggregated data. Samples above 20 mg C L^{-1} are not included in the graph for visual clarity ($n = 337$). The black dashed lines indicate the global median (1.2 mg C L^{-1}) and mean (3.8 mg C L^{-1}) and 95th percentile (16.6 mg C L^{-1}) values respectively. Also shown are the 99th percentile value and the maximum value (33.0 and $1040.0 \text{ mg C L}^{-1}$ respectively). B) countries from which groundwater DOC data was obtained.

We have also included the following new text:

Line 86 – 88: “Most groundwater DOC concentrations fall within the $0 - 5 \text{ mg C L}^{-1}$ range, with 84.1% of samples $< 5 \text{ mg C L}^{-1}$ (Fig 1A), with the dataset dominated by countries in low and mid latitudes”

Reviewer #2 (Remarks to the Author):

I think the authors should be congratulated on an excellent set of well-argued responses, thoughtful revisions, and a final manuscript that I think is an excellent contribution. I was not entirely convinced by the earlier drafts of this manuscript, but I feel the latest additions are a significant improvement. Thank you for taking the time to address the comments in such a thorough way.

I think the paper should be published in its current form.

We thank the reviewer for their positive feedback and are happy that the revisions adequately addressed all the reviewer's comments.

Reviewer #3 (Remarks to the Author):

I focus my review on the new figure added to this version of the paper and to the two suggestions I made in my previous review. In my opinion, all three of these issues must be addressed.

New Figure.

I agree that the new figure helps to convey the conclusions of the paper. However, this figure greatly exaggerates the magnitude of the results. For example, panel B shows that DOC concentrations are double in urban than in natural areas. I added a red line showing the actual finding of the paper, a 19% difference. The red line accurately represents the finding of the paper while the original black line exaggerates the finding. It's important to understand that the difference between natural and urban locations found in this paper was a modest 19% to place the result in context of the variability of DOC concentrations which is often orders of magnitude. Some of the other panels also appear strongly exaggerate the findings: the lines representing DOC should be much closer to horizontal in panel A. Yes, the p-values that the statistical significance of the findings is strong. However, the effects of climate are modest whereas the figure implies, incorrectly, that DOC concentrations go to zero at the origin or on the right side of the graph.

We note this concern and have revised the figure as suggested below:

Fig. 3. Conceptual model summarizing the drivers of change in groundwater DOC concentrations. (A) Climate parameters: in arid climates, groundwater DOC concentrations increase with increased precipitation due to the priming of organic matter by microbes under warm and increasingly wet conditions. Increased temperatures in arid environments reduced groundwater DOC due to increasing aridity. Precipitation in humid environments decreases groundwater DOC concentrations due to dilution whilst temperatures increase DOM priming by microbes. (B) Urban land use contributes to groundwater DOC through contamination, for example through leaking septic and sewer systems. (C) Water quality parameters and groundwater DOC concentrations are linked and are largely controlled by redox conditions (NB: C shows variables where DOC is the driver for the observed changes in water chemistry). (D) Aquifer age results in a decline in groundwater DOC due to sediment lithification and a depletion of sedimentary organic matter over time. Longer groundwater residence times decreased DOC by a combination of filtration of DOC through smaller aquifer pore sizes and adsorption (where residence times correspond to longer flow paths), and increased exposure to biodegradation over time.

I agree with the authors that this figure is useful because it is important to visually convey the direction of the effects, but is equally important to convey the magnitude of the effects. By adding values to the X-axes and adjusting the lines to represent the actual results, this figure can convey the findings, including the important finding that many of the effects are quite modest. I understand that this figure is meant as a “cartoon,” not as quantitative results. None-the-less, this figure will mislead readers as it is.

We agree and feel that the revised figure shown above now much more accurately represents the magnitude of the effects of the factors on groundwater DOC.

Previous Comment #1. This paper concludes that urbanization causes DOC concentrations to rise. This is why analysis of DOC data before and after urbanization is important. However, the result rests entirely on a “space for time” analysis, comparing concentrations across urban and natural sites. The problem with this analysis is that it cannot prove that DOC in natural areas will rise if those areas are urbanized. There could be other explanations for the observed 19% difference. For example, it could easily be that urban development has tended to be in low-lying areas near bodies of water where there are more wetlands that are a source of DOC. This seems a very plausible explanation for a 19% difference to me. I appreciate that the authors worked hard to find time series of DOC data. In their response to my comment they, they describe how their analysis of data from Perth and Florida showed “a mix of trends including positive, negative or no trend which further confirmed the that lack of relationship between DOC concentration and time urban areas.” Why did they not include this in their paper?

We have now included the timeseries data in the supplementary (Fig. S5) showing the Perth and Florida data with loess fits for each bore showing a lack of clear trend in the data.

Fig. S5.

Timeseries of groundwater TOC concentrations in south-west Florida, United States (upper left plot) with corresponding histogram shown on the upper right. Timeseries of groundwater DOC concentration data in Perth, Australia (lower plot) with corresponding histogram shown on the lower right). TOC data used for Florida due to the paucity of groundwater DOC datasets available. Here we assume that majority of the TOC in groundwater is dissolved. Both datasets represent currently residential areas. Grey dots represent individual concentration data with dashed lines representing locally estimated scatterplot smoothing (LOESS) colored by correlation coefficient for individual bores ($n = 45$ bores and $n = 51$ bores for Perth and Florida, respectively). LOESS smoothing used as many datasets are non-linear. The data suggests a mix of trends including increasing concentrations, decreasing concentrations and no change in concentrations over time. Florida and Perth data were provided by the Southwest Florida Water Management District and Water Corporation (Western Australia) respectively.

As noted previously, we have also added the following text to the manuscript:

Line 256 – 258: “Our finding of increased DOC in urban areas from a space-for-time analysis, as well as a previous space-for-time analysis⁵⁵, cannot reveal how this increase has occurred over time. A search for available groundwater TOC and DOC timeseries data in

urban areas produced two datasets from Florida, US, and Perth, Australia. These data, collected from the 1980's to present in largely-residential urban areas, show no clear trend in groundwater DOC (Fig. S5) over this timescale. Longer groundwater DOC timeseries on time scales longer than aquifer residence times would be needed to confirm our space-for-time interpretation. For example, fluvial DOC concentrations in the Thames Basin since 1883⁵⁶ have increased since World War 2 due to sewerage inputs and changes in land management. Further groundwater DOC time series observations are required to assess the impact of urban area expansion, for example into lowland regions where DOC might be high, mobilization of previously stable soil DOC following development, and legacy contamination of groundwater in urban areas.”

The paper needs to give an objective analysis of the question of causation versus correlation. The authors argue that they do not have enough time-series data to answer the question, and chose not to include the only data they could find. As the paper reads now, many readers will think that the authors really did determine that DOC concentrations increase when land is urbanized. To avoid misleading readers, the paper must state up-front and prominently that they do not analyze any data describing how DOC concentrations change before and after urbanization. Rather, they perform a space-for-time analysis and the results of this are consistent with such a change, but are also consistent with other explanations. The result that DOC is higher in urban groundwater is useful, but should not be confused with the idea that urbanization causes DOC to increase.

We note our response to this in the above text. We have also added an additional note in the implications:

Line 295 – 298: “Our analysis suggest that this could lead to increased groundwater DOC concentrations, however these results are based on a space-for-time analysis. DOC time series in groundwater over time scales longer than aquifer residence times are required to confirm our model.”

Previous Comment #2. The authors need to go further in addressing this comment. To explain, I'll use the cases of Senegal and Nigeria. Senegal is shown with the highest DOC concentrations and Nigeria is shown with the eighth from the lowest. DOC appears to be about seven times higher in Senegal than Nigeria and the standard deviations for the two data sets do not overlap. Of course, to someone who has experience in this field, it is immediately obvious that this data cannot be used to determine that Senegal has higher DOC concentrations because there was no effort to design a sampling scheme to find a national average in either country and their a very few data points in both countries. **But to other readers this will not be obvious and there can be real consequences.** For example, it is conceivable that an NGO could read this paper and conclude that a water treatment program should all focused in Senegal without any concern for Nigeria. I would recommend replacing these box-and-whisker plots with different categories. Instead of countries, show categories for which the authors believe there is a real difference in DOC, such as age of aquifer or depth. If the authors insist on keeping the current plots, that differentiate the data by country, then they should clearly and prominently state that this plot cannot be used to determine country by country levels of contamination — even though that's what it appears to show.

We agree that the country-scale analysis could be easily misinterpreted. We have modified Fig. 1 to show an aggregate histogram of all groundwater DOC concentrations available and have avoided separating DOC concentrations by country in order to prevent any potential misinterpretation by a reader. We refer to our response to Reviewer 1's second comment.

We thank the reviewers and editor for their comments. END OF REVIEWS

REVIEWERS' COMMENTS:

Reviewer #1 (Remarks to the Author):

I thank the authors for devoting substantial time and effort to revising their manuscript, including this latest round of revisions that addressed my earlier comments.

Reviewer #3 (Remarks to the Author):

The authors addressed all of my concerns.

Changes in global groundwater organic carbon driven by climate change and urbanization

Response to review comments received February 1, 2020.

Reviewer Comments:

Reviewer #1 (Remarks to the Author):

I thank the authors for devoting substantial time and effort to revising their manuscript, including this latest round of revisions that addressed my earlier comments.

We thank the reviewer for their positive feedback.

Reviewer #3 (Remarks to the Author):

The authors addressed all of my concerns.

We thank the reviewer for their positive feedback.

END OF REVIEWS